# Correlation of Perfusion Metrics with Ki-67 Proliferation Index and Axillary Involvement as a Prognostic Marker in Breast Carcinoma Cases: A Dynamic Contrast-Enhanced Perfusion MRI Study

**DOI:** 10.3390/diagnostics13203260

**Published:** 2023-10-19

**Authors:** Ulas Yalim Uncu, Sibel Aydin Aksu

**Affiliations:** 1Department of Radiology, Van Training and Research Hospital, University of Health Sciences, 65300 Van, Turkey; 2Department of Radiology, Haydarpasa Numune Training and Research Hospital, University of Health Sciences, 34668 Istanbul, Turkey; drsibelaydin@yahoo.com

**Keywords:** perfusion MRI, breast cancer, prognosis, axillary stage

## Abstract

Our study aims to reveal clinically helpful prognostic markers using quantitative radiologic data from perfusion magnetic resonance imaging for patients with locally advanced carcinoma, using the Ki-67 index as a surrogate. Patients who received a breast cancer diagnosis and had undergone dynamic contrast-enhanced magnetic resonance imaging of the breast for pretreatment evaluation and follow-up were searched retrospectively. We evaluated the MRI studies for perfusion parameters and various categories and compared them to the Ki-67 index. Axillary involvement was categorized as low (N0–N1) or high (N2–N3) according to clinical stage. A total sum of 60 patients’ data was included in this study. Perfusion parameters and Ki-67 showed a significant correlation with the transfer constant (K^trans^) (ρ = 0.554 *p* = 0.00), reverse transfer constant (K^ep^) (ρ = 0.454 *p* = 0.00), and initial area under the gadolinium curve (IAUGC) (ρ = 0.619 *p* = 0.00). The IAUGC was also significantly different between axillary stage groups (Z = 2.478 *p* = 0.013). Outside of our primary hypothesis, associations between axillary stage and contrast enhancement (x^2^ = 8.023 *p* = 0.046) and filling patterns (x^2^ = 8.751 *p* = 0.013) were detected. In conclusion, these parameters are potential prognostic markers in patients with moderate Ki-67 indices, such as those in our study group. The relationship between axillary status and perfusion parameters also has the potential to determine patients who would benefit from limited axillary dissection.

## 1. Introduction

Breast cancer is the most common cancer type in women and the second most common cause of cancer-related death [1]. Early diagnosis and effective treatment markedly improve survival rates. However, it is a heterogeneous disease with different morphologic features, clinical courses, and treatment responses because of tumors’ distinct genomic and immunological diversity. Therefore, personalized management is required. This is particularly true for patients receiving neoadjuvant chemotherapy (NAC), which creates a different challenge for decision making. As part of the patient’s care routine, molecular markers have been added to traditional parameters, such as tumor size, histological grade, and lymph node status [2]. Due to its high cost and time requirements, gene expression profiling is not widely applied to breast cancer subtyping, even though it is the gold standard. The St. Galen consensus is essentially an immunological surrogate for molecular subtypes of tumors [3]. A vital component of this classification is the Ki-67 index, a valuable marker for distinguishing luminal A and luminal B breast cancers [4].

Ki-67 protein is an immunohistochemical marker for cell proliferation and is exclusively expressed during the metaphase of the cell cycle. The Ki-67 proliferation index, determined by the percentage of positive staining cells for Ki-67 protein, is used as a prognostic marker in breast cancer [3]. As a measure of the extent of proliferative activity, it is an effective method of identifying more aggressive breast cancers [3].

Breast MRI is an imaging modality considered the gold standard in breast imaging with high sensitivity and specificity; in invasive tumors, they are even higher. A recent study reported a 95% confidence interval of 96.5–97.6% for specificity with a positive predictive value of 35.7% in diagnosing high-grade breast tumors of sizes as small as 8 mm [5]. Apart from diagnostic workup, dynamic contrast-enhanced magnetic resonance imaging (DCE-MRI) is also a powerful tool for staging local disease in newly diagnosed breast cancer and monitoring treatment during NAC [6].

Perfusion is a physiological parameter representing a unit of blood flow delivered to a given amount of tissue (mL/s/100 g) [7]. MRI can be used to measure tissue perfusion with a calculation of gadolinium concentration in the tissue over time. Gadolinium-based contrast has a paramagnetic property that manifests as T1 shortening in MRI. Using this property, it is possible to quantify the amount of Gadolinium in a voxel [8]. Dynamic contrast-enhanced perfusion magnetic resonance imaging (DCE-perf MRI) has the advantage of measuring contrast delivery to tissues and passage of contrast to extracellular tissues. DCE-MRI-derived parameters, including the transfer constant (K^trans^), reverse transfer constant (K^ep^), and extracellular volume (V_e_), are quantitative measures of perfusion reflecting the extent of tumor angiogenesis. K^trans^ represents the combination of contrast delivery and tissue permeability and has been suggested for tumor surveillance [9]. Several studies successfully showed K^trans^ in evaluation response and treatment follow-up in breast cancer. This topic is also well researched in the follow-up of brain tumors [10,11,12]. In addition, the model procures additional parameters such as the initial area under the gadolinium curve (IAUGC), which is the area under the gadolinium time-concentration curve; V_e_, which is necessary for quantifying the contrast concentration accurately; and K^ep^, which reflects the rate at which the contrast material returns from the extracellular space and into the venous system [8].

There is a recent trend toward foregoing axillary dissection in patients who receive neoadjuvant chemotherapy. The decision is based on the initial level of axillary involvement, pathologic response to treatment, and surgical considerations [13]. The decision is partly based on initial evaluation, as targeted axillary dissection requires pre-treatment marker placement, and the surgical specimen is only available after treatment is completed.

The objective of our study was to explore the possibility of extracting additional prognostic information from the initial imaging studies of breast cancer patients. Specifically, we employed the Ki-67 index as a surrogate marker for tumor aggressiveness, aiming to determine whether perfusion parameters show a correlation with higher Ki-67 indices. The parameters we identified in our study could be included in future investigations as potential independent prognostic markers. Additionally, given the relevance of axillary involvement to surgical decision making, we sought to explore correlations between extensive axillary involvement and perfusion. Given that quantitative DCE perfusion imaging is not a standard practice in breast MRI, we also included several categorical imaging variables as secondary parameters.

## 2. Materials and Methods

This retrospective study was conducted in accordance with the Declaration of Helsinki and approved by the Academic Review Board of Health Sciences University, per the procedure for thesis dissertations (date: 12 October 2020, number: 48865165-302.14.0135414).

This study includes patients diagnosed with primary breast carcinoma in our institution between July 2019 and June 2020 and who had a pretreatment breast MRI study performed. Among these, only patients with a quantitative DCE MRI sequence were included; this sequence was only used for patients who would receive neoadjuvant chemotherapy. Patients who did not have a pathology report in our hospital; patients diagnosed with breast diseases other than primary ductal or lobular carcinoma of the breast; and patients who received surgery, chemotherapy, or radiotherapy before imaging were not included in this study.

Our study examined the cases with a 1.5 T MRI scanner (GEGE Optima 36 Bamboo; General Electric, Milwaukee, WI, USA). Imaging was performed with a dedicated 16-channel phased-array bilateral breast coil in the prone position. An intravenous bolus injection of 0.1 mmol/kg Gadobutrol (Gadovist; Bayer Schering Pharma, Berlin, Germany) was administered at 2 mL/s, followed by a 20 mL saline flush. The images were obtained with the minimum FOV which includes complete breast tissue bilaterally, level one lymph nodes in the axilla, and both claviculae.

Pre- and post-contrast imaging was performed using the VIBRANT sequence with the following parameters: TETE: 2.9 TR: 6, flip angle 10 degrees, and matrix 384 × 256. In addition, an early phase (injection of contrast was performed after the first three baseline images) sequence with similar weighting but lower resolution and fewer slices but high temporal resolution was performed to quantify the perfusion kinetics of lesions. A moderate temporal resolution was used as proposed by certain studies [14]. Motion correction was applied using the post-processing tools of the GEGE Advantage Workstation as needed. We assumed a fixed T1 in our model as pre-contrast T1 time determination was unavailable. Evidence in the literature suggests that the results are comparable [15].

As our study was performed retrospectively, all images were already reported by a board-certified radiologist with five years of experience in breast imaging. However, the reports did not include all the quantitative data, so we re-evaluated the images for this study.

According to the Breast Imaging Reporting and Data System (BIRADS) 5th edition manual, the contrast enhancement pattern was classified as homogenous, heterogenous, and peripherally enhancing. Heterogeneously enhancing tumors were subdivided into tumors that include non-enhancing voxels and those that show complete but heterogeneous enhancement. Post-contrast images at approximately five minutes after injection were used to characterize these patterns.

Perfusion imaging was analyzed using a manual arterial input function. The right internal mammary artery (RIMA) was used as a standard, and three voxels were averaged. Voxels that enhanced above 20% of others were not included to avoid artifacts due to arterial motion. The LIMA was referenced for quality assurance.

After the arterial input function configuration, K^trans^, V^e^, K^ep^, and IAUGC maps were generated. A ROI was placed on three consecutive slices where the enhancing portion of the tumor was the largest. A ROI was hand-drawn along the enhancing part of the tumor. In the case of marked enhancement heterogeneity in large tumors, an area of consistent enhancement with a minimum size of 10 cm^2^ was selected.

In addition to perfusion maps, a contrast filling pattern was ranked according to time to peak enhancement as slow filling (>20 s), fast filling (≤20 s) and an additional rank, rapid washout at the periphery.

The imaging protocol was reserved for patients with locally advanced breast cancer who would receive NAC and axillary involvement was common. We used clinical axillary staging, supported with radiologic evidence (in either ultrasound, MRI, or 18-deoxyflouro-FDG PET studies) of over four suspicious lymph nodes to determine the N2 stage. Although the AJCC does not recommend routine radiologic staging based on the affected lymph node number [16], this approach is supported in the literature. Our institution commonly uses this criterion to determine candidates for limited axillary excision [17]. AJCC parameters were used to determine the N3 stage [16], and for this study, N2–N3 were combined into the “high lymph node stage”.

All patients included were diagnosed with a core needle biopsy. A pretreatment surgical excision specimen was never available. Results included tumor subtypes according to the St. Galen consensus. The Ki-67 index was routinely reported according to the percentage of cells in the area with the highest staining [18].

Group comparisons were performed using Student’s *t*-test or the Mann–Whitney U test based on the normality of the data. Spearmen’s rank correlation was used to test the correlation between variables. Categorical data were analyzed using the Chi-Square test for two binary categories and the Kendall–Tau C test for categorical data with multiple ranks (such as contrast filling and contrast enhancement). Statistical significance was determined as *p* < 0.05.

## 3. Results

Seventy-one patients with DCE-perfusion MRI were found in our records; however, eight patients were removed based on our exclusion criteria. Among the 63 patients who met the selection criteria, 3 patients had to be excluded due to a lack of image quality in the dynamic sequence because of the motion.

Several examples of cases are shown in Figure 1, Figure 2 and Figure 3.

All 60 patients were female. Patient age showed normal distribution with a mean age of 46.25 years and a standard deviation of 11.65 years. The tumor subtype distribution is shown in Figure 4.

The tumor size in three orthogonal planes was measured and averaged. The average tumor dimension showed a non-normal distribution; the median value was 28.55 mm, and the interquartile range was 28.95 mm. The longest tumor diameter showed a median value of 39.5 mm with an interquartile range of 40.

In a comparison of the enhancement pattern ranks with the presence of tumor necrosis, the Kendall–Tau-C test was used. A significant relationship was found between the contrast enhancement pattern (τ = 0.536, *p* = 0.000) and the contrast filling pattern (τ = 0.309 *p* = 0.037). Peripheral enhancement and rapid washout showed a strong association with tumor necrosis in their respective categories. When tumor necrosis was compared to continuous variables, the Mann–Whitney U test was applied, and significant differences were found with regard to K^trans^, Ki-67, and IAUGC parameters (*p* = 0.011, *p* = 0.003 *p* = 0.034, respectively). According to Shapiro–Wilk tests, K^trans^, K^ep^, V_e_, and IAUGC values showed a non-normal distribution. Values are demonstrated in Table 1. When perfusion parameters were compared to tumor size using Spearmen’s rank correlation, the anteroposterior tumor diameter was associated with K^trans^ (ρ = 0.355 *p* = 0.005) and K^ep^ (ρ = 0.310 *p* = 0.016). Other size parameters were not associated with perfusion parameters. Our study was not predicted to have statistical power to differentiate between immunological tumor subtypes due to lower case numbers in some groups, and the difference in perfusion parameters between the groups was not statistically significant. However, Her-2/Neu positivity, as an independent marker, was significantly associated with K^ep^ (Z = 2.069, *p* = 0.039).

Ki-67 indices showed a non-normal distribution; the median value was 36.5%, and the interquartile range was 37%. When comparing continuous variables, Ki-67 was not associated with patient age. The Ki-67 index showed a correlation with multiple perfusion parameters, K^trans^ (ρ = 0.554 *p* = 0.00) and K^ep^ (ρ = 0.454 *p* = 0.00); IAUGC (ρ = 0.619 *p* = 0.00) but did not show a relationship with the V_e_ parameter (Figure 5). In addition, Ki-67 showed a positive relationship with both contrast enhancement and contrast filling ranks (ρ = 0.338 *p* = 0.008 and ρ = 0.268 *p* = 0.039) in rank correlation analysis.

Regarding the lymph node stage, 31 cases were in the N1–N2 group and 29 were in the high lymph node stage (N2–N3) group. Axillary stage groups were compared with other categorical variables. The contrast enhancement pattern was significantly correlated with the axillary stage with the Chi-Square test (x^2^ = 8.023 *p* = 0.046) and the Kendall–Tau C test (τ = 0.388 *p* = 0.003). Tumor enhancement and filling pattern percentages, according to axillary subgroups, are referenced in Table 2. Her-2/Neu positivity was associated with a high axillary stage (x^2^ = 6.901 *p* = 0.009). The contrast filling pattern was associated with the axillary stage (x^2^ = 8.751 *p* = 0.013). The axillary stage was correlated with IAUGC values (Mann–Whitney U test Z = 2.478 *p* = 0.013); the correlation with other perfusion parameters is listed in Table 3.

## 4. Discussion

In contemporary breast oncology practice, there is a greater focus on minimally invasive procedures and increased use of NAC and adjuvant axillary radiotherapy. Unlike surgical treatments, radiotherapy and chemotherapy do not procure an immediate pathology result, and there is great outcome variability depending on patient and tumor factors. For these reasons, breast MRI is the gold standard for patient follow-up [19,20].

An MRI-based prognostic classification would be valuable in patients receiving NAC for two reasons. First, MRI is the golden standard to monitor the treatment progress and, therefore, availability and cost are minor issues, as many patients would get an MRI regardless. Secondly, treatment decision is primarily based on core biopsy samples, and although reporting is standardized worldwide, re-classification after resection is possible. As these patients must receive a long and cost-incurring treatment process, any additional data that can mitigate error early in the approach would be valuable [21]. Our study aims to evaluate the possible role of MRI perfusion as a prognostic tool and search for possible quantifiable data related to current known pathologic prognostic factors. Our study has a retrospective design and consists of patients selected for NAC; therefore, the population does not represent all breast cancer patients, especially concerning the most common subtype of Luminal A.

Tumor necrosis showed significant associations with multiple radiologic factors. The enhancement and tumor necrosis pattern showed a prominent relationship; peripherally enhancing tumors and tumors with non-enhancing voxels more frequently display pathologically evident tumor necrosis than homogeneous and mildly heterogeneous tumors. The value of tumor heterogeneity is described in many studies [12,22,23], which may explain the observation in our study. These findings are also supported by experimental studies on tumor xenografts that demonstrated enhancement patterns and their relation to tumor necrosis [24,25]. The presence of tumor necrosis is also correlated with Ki-67, K^trans^, and IAUGC. Conversely, Abdelhafez et al. (2021) argued against the prognostic value of apparent necrosis in pretreatment MRI [26]. However, tumor heterogeneity is only a secondary parameter in our study, and reliance on core biopsy limits the histopathological detection of necrosis. In our opinion, further studies with quantifiable markers of vascularity and necrosis in pathological examination are necessary to explore the implications of this finding and its relation to tumor heterogeneity. Wu et al. (2018) demonstrated quantitative tumor heterogeneity as an independent prognostic factor. Nevertheless, its association with histological tumor necrosis should also be tested in a study with independent analysis of each factor in the patient prognosis [23].

Other studies have shown success in differentiating immunological subtypes of breast cancer [26]. However, our study lacked statistical power to demonstrate this association due to low numbers of some tumor subtypes. We found Her-2/Neu positivity to be associated with K^ep^, but the significance of this isolated finding outside our hypothesis is unclear.

The primary hypothesis of our study is that there is a potentially useful association between DCE-perf MRI parameters and the Ki-67 proliferation index.

Our study revealed a significant relationship between several perfusion parameters and the Ki-67 index. The IAUGC showed the strongest correlation (ρ = 0.619 *p* < 0.01), followed by K^trans^ (ρ = 0.554 *p* < 0.01) and K^ep^ (ρ = 0.454 *p* < 0.01). Shin et al. (2017) also demonstrated that the Ki-67 index was strongly correlated with K^trans^ values with a cut-off of 0.274 [27]. Kim et al. (2016) demonstrated that K^trans^ is associated with the histological evaluation of vessel density [28]. The proposal of these studies, similar to ours, is that perfusion parameters are a surrogate for tumor angiogenesis and, by proxy, tumor aggressiveness.

Our study has not shown a significant correlation between V_e_ and other parameters. V_e_ is an expression of extracellular volume and is inherently negatively associated with cellularity. Nagasaka et al. (2019) showed that V_e_ is strongly correlated with Ki-67, and this finding is not reflected in our study [29]. Kang et al. (2020), like our study, did not find V_e_ to be associated with the Ki-67 index [30]. This may be because both studies did not use a pre-bolus determination of T1 values, as in the Tofts model correct pre-bolus T1 is used to determine absolute contrast enhancement per voxel and, therefore, V_e_. Additionally, Thakran et al. (2018) did not demonstrate a relationship between V_e_ in contrast to other perfusion parameters [31]. However, they attribute parts of their results that do not fit within the perfusion model to using a standardized arterial input function.

In addition to quantitative parameters, Ki-67 showed a positive relationship with both contrast enhancement and contrast filling ranks; we believe such parameters should be included in research on tumor kinetics for several reasons. Though quantifiable parameters spark much academic interest and allow for different statistical analyses, there is the issue of replicability. Combining the results from different studies is difficult since the measurements depend on the exact imaging parameters and software models. Because of these reasons, we believe non-quantifiable parameters have a place and can be clinically helpful based on the setting.

The patients in our study were also grouped for their axillary status as low (N0–N1) and high axillary stage (N2–N3) patients, per our methods section. When these two groups were compared, there was a significant difference in the IAUGC regarding axillary groups. Loiselle et al. (2014) demonstrated that DCE-MRI findings can improve the diagnostic value of the Katz nomogram. Their model uses multiple DCE-MRI parameters, and the percentage of persistent enhancement was shown to be of value [32]. Our study used a different quantification method of enhancement. However, there is discordance in that they demonstrated persistent enhancement to a more significant predictor of axillary involvement compared to rapid filling, whereas our finding of rapid filling (≤20 s to peak) and washout correlated with axillary status with a stronger association at each rank. The difference in temporal resolution may explain the discordant results, as ultrafast DCE-MRI has shown that temporal resolution can yield additional diagnostic information [33].

Further exploring the concept of fast-filling and ultrafast DCE-MRI, Yamaguchi et al. (2021) showed it correlated with axillary metastasis. However, to our knowledge, a study that investigated ultrafast MRI or DCE-perfusion specifically with the extent of axillary involvement has yet to be performed [34]. Ya et al. (2022) demonstrated the K^trans^, K^ep^, and IAUGC (AUC in their study), with the AUC having the most predictive value in their model. Our study differs from theirs because we predicted the presence of greater than four metastatic lymph nodes, as opposed to complete absence [12]. Han et al. (2019) demonstrated that a radiomics-based model (without quantitative perfusion) can predict greater than or less than two axillary lymph nodes [35]. Additional studies for DCE-MRI’s prediction of axillary involvement, focusing on patients who could receive limited axillary dissection and their long-term axillary recurrence, would be beneficial. The Ki-67 index did not show a significant relationship between high and low axillary groups. This finding is exciting as it may point to tumor vascularity and perfusion parameters as independent indicators for extensive axillary involvement.

It should be noted that our neoadjuvant patient group was in a limited range of Ki-67 indices, as many patients with low Ki-67 indices do not receive neoadjuvant chemotherapy. Although our study results cannot be generalized to all breast cancers, these findings may be a marker for guiding chemotherapy and axillary surgical approaches in patients with moderate to high Ki-67 indices. This is particularly interesting for patients who receive neoadjuvant chemotherapy and could benefit from a limited axillary dissection.

Our study has several important limitations. First, owing to the retrospective nature of our study, we used patients who received this MRI protocol for follow-up during neoadjuvant chemotherapy. This means that the study does not include any breast cancer patients who went to primary surgery or patients with metastatic disease who would receive primary chemotherapy; therefore, our population is inherently biased. Our study draws conclusions from indirect associations between known pathological and radiological prognostic factors. It should be considered a hypothesis-generating study, and further studies that directly compare survival and patient outcomes to these parameters are needed.

The second significant limitation is that the procedure for quantitative MRI is a limited protocol designed to make quantitative comparison possible and did not include a T1 mapping that would be necessary to standardize our findings with other research. Finally, our patient number was not adequate to test the accuracy of our parameters individually for each tumor subtype, and all breast cancers were analyzed as one group.

## 5. Conclusions

In conclusion, we believe that quantifiable and categorical data with significant prognostic value can be gathered from pretreatment DCE-MRI studies of breast cancer patients. Our study demonstrates a significant linear correlation between IAUGC, K^trans^ K^ep^ values, and the Ki-67 index. In addition, extensive axillary involvement (four or more axillary lymph nodes) is associated with perfusion parameters. There is extensive research on perfusion parameters and how they relate to breast cancer subgroups; however, we believe that more research focusing on the axillary status of locally aggressive breast cancer patients is necessary to investigate the potential for diagnostic decision making based on perfusion parameters.

## Figures and Tables

**Figure 1 diagnostics-13-03260-f001:**
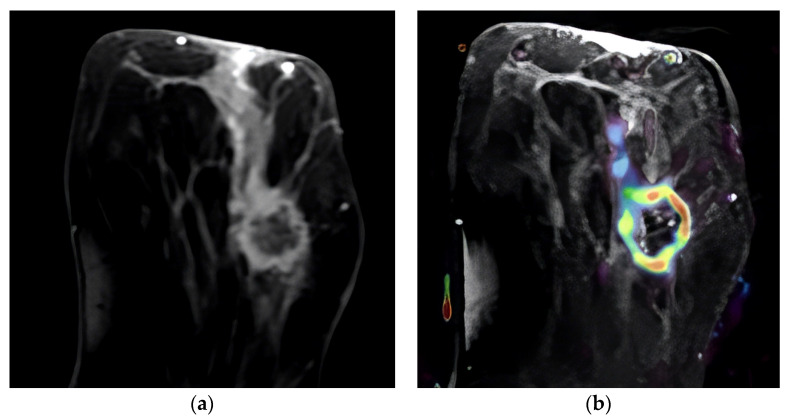
Ring-enhancing mass in the left breast. A HER-2-positive tumor, Ki-67 index: 45%. (**a**) Post-contrast image of the tumor demonstrating ring enhancement with a linear enhancing component; (**b**) overlay of the same image with IAUGC perfusion map, red denotes high perfusion.

**Figure 2 diagnostics-13-03260-f002:**
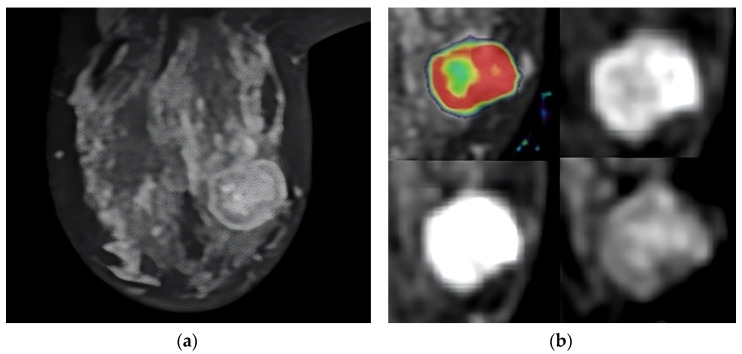
T2-weighted MRI image that shows the tumor’s location in the axial slice (**a**). Perfusion mapping (red shows elevated perfusion) and progressive contrast change show washout in the tumor’s periphery (**b**).

**Figure 3 diagnostics-13-03260-f003:**
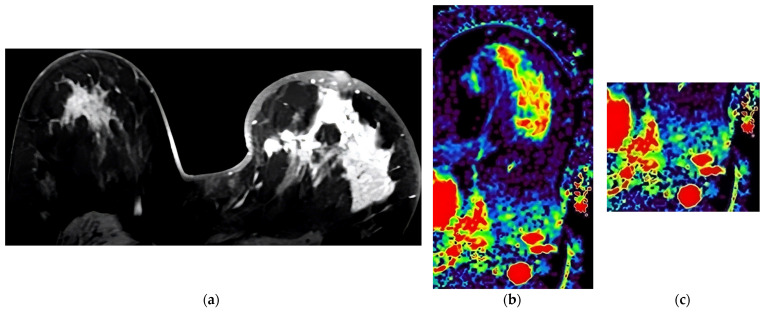
Case presented with diffuse heterogeneous-enhancing Luminal B tumor in the left breast with high Ki-67 proliferation index (67%). (**a**) Last phase post-contrast image of the tumor. (**b**) IAUGC map of the tumor at the same level (red color denotes elevated perfusion). (**c**) IAUGC map showing spherical axillary lymph nodes with increased perfusion.

**Figure 4 diagnostics-13-03260-f004:**
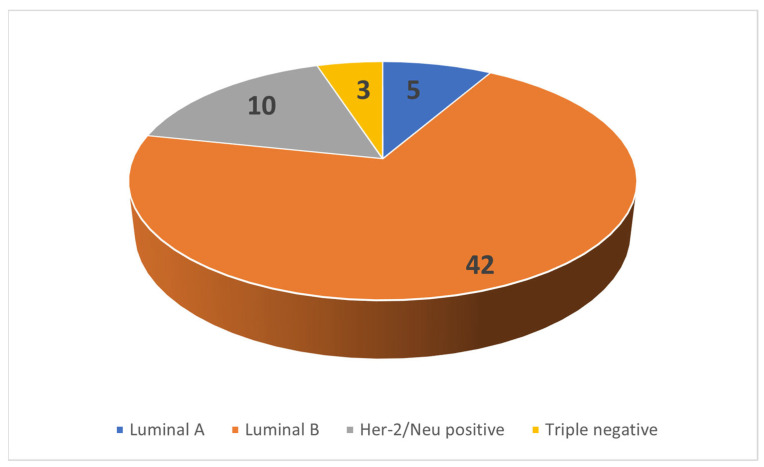
Distribution of cases.

**Figure 5 diagnostics-13-03260-f005:**
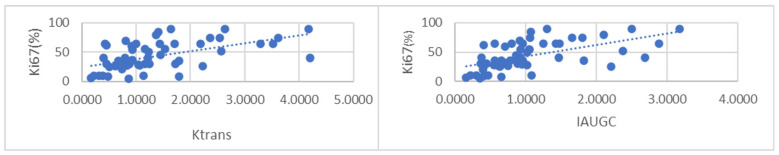
Scatterplots of K^trans^ (**left**) and IAUGC (**right**) with Ki-67 index.

**Table 1 diagnostics-13-03260-t001:** Perfusion parameters.

Perfusion Parameter	N	Mean	Median	IQR
K^trans^	60	1.320000	0.966	0.976
K^ep^	60	1.96800	1.412	1.624
V_e_	60	0.771000	0.834	0.314
IAUGC	60	1.034000	0.906	0.641

N: number, IQR: interquartile range.

**Table 2 diagnostics-13-03260-t002:** Tumor enhancement and filling patterns, and distribution of axillary stage.

Enhancement Pattern		Number (% of Total)	Low Axillary Stage (%within Pattern)	High Axillary Stage (%within Pattern)
Homogenous		17 (28.3)	13 (76.5)	4 (23.5)
Heterogeneous	Complete enhancement	23 (38.3)	12 (52.2)	11 (47.8)
	Non-enhancing voxels	11 (18.3)	3 (27.3)	8 (72.7)
Peripheral		9 (15)	3 (33.3)	6 (66.7)
Total		60	31 (51.7)	29 (48.3)
Filling Pattern				
Slow filling		39 (65)	24 (61.5)	15 (38.5)
Rapid filling		8 (13.3)	5 (62.5)	3 (37.5)
Rapid washout		13 (21.7)	2 (15.4)	11 (24.6)
Total		60	100	

**Table 3 diagnostics-13-03260-t003:** Perfusion parameters according to LNLN stage groups.

Perfusion Parameter	LN Stage	N	Mean	Median	IQR	25–75 Percentiles	AUC	*p* Value *	Z Score *
K^trans^	Low (N0–1)	31	1.173065	0.832	0.913	0.614–1.527	0.642	0.059	1.886
	High (N2–3)	29	1.478069	1.129	0.957	0.834–1.792			
K^ep^	Low (N0–1)	31	1.71285	1.175	1.852	0.848–2.700	0.630	0.085	1.723
	High (N2–3)	29	2.24086	1.485	1.437	1.234–2.676			
V_e_	Low (N0–1)	31	0.749968	0.791	0.288	0.618–0.906	0.618	0.115	1.575
	High (N2–3)	29	0.793252	0.909	0.348	0.632–0.980			
IAUGC	Low (N0–1)	31	0.841161	0.654	0.657	0.408–1.070	0.686	0.013	2.478
	High (N2–3)	29	1.241069	0.985	0.893	0.761–1.654			

LN: lymph node, N: number, IQR: interquartile range, AUC: area under curve (ROC), * statistically significant (*p* < 0.05, and Z score > 1.95).

## Data Availability

The data presented in this study are available upon request from the corresponding author. The institutional board approved the use of data by researchers for this research, but uploading patient information to a public database was not included in this approval.

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
