# Peer review of "Correlation of Perfusion Metrics with Ki-67 Proliferation Index and Axillary Involvement as a Prognostic Marker in Breast Carcinoma Cases: A Dynamic Contrast-Enhanced Perfusion MRI Study"

_diagnostics, 2023, doi:10.3390/diagnostics13203260_

Round 1

Reviewer 1 Report

The present article was very difficult to read because the aim, the material and methods, the statistical analysis and the results of the study, are not clearly reported.

From the title, the aim of this study should be to explore the correlation between Perfusion Metrics and Ki-67 Index 2 and Axillary Involvement. 

However, reading the paper, it emerges that the authors perform a statistical correlation analysis between some perfusion parameters derived from MRI and compare them with a series of clinical-bio-histological parameters, finding the aforementioned correlation.

This methodology of research it is not strong enough to deduce scientific evidence.

Extensive editing of English language required

Author Response

Thank you for your review, please see the attachment for our response.

Reviewer 2 Report

The authors report on the correlation of perfusion metrics with Ki-67 proliferation index and axillary involvement as a prognostic marker in breast carcinoma: a dynamic contrast-enhanced perfusion MRI study.

Comments

Indeed, breast cancer is the most frequent malignancy in women.

A correct imagistic diagnosis is fundamental for an efficient further therapy and, unfortunately, many times this desiderate remains unaccomplished, despise the more and more sophisticated techniques.

I appreciate that the present study is well structured and monitored.

Also, I consider that the dynamic contrast-enhanced perfusion MRI is of real interest for the reader and constitute a promising method for a more accurate diagnostic of this terrible disease.

I recommend to be published in the present form.

Good work!

Author Response

Thank you for taking the time to review our manuscript, and your kind feedback.

Reviewer 3 Report

Major remarks:

Add some information about gadolinium contrast in Introduction, please.

Add an Enhancement and Filling Patterns in Table 3.

Minor remarks:

Add spaces between numbers and "=", please.

Change commas to dots in numbers, please (e.g. line 23: "Z=2,478 p=0,013" => Z = 2.478, p = 0.013).

Line 75: Add space before citation, please ("returns from the extracellular space and into the venous system[8]." => ... system [8].).

Line 133: add space in the sentence: "... a minimum size of 10 cm2was selected.", please.

I suggest move Figures 1-3  to Results section.

Line 213: "Her2Neu" => Her-2/neu

Lines 249, 252, 275, 276, 295, 304, 307, 310: Correct citations (e.g. Surname et al. (year) ... citation in the end of sentence).

Line 257: Move citation to the end of the sentence, please.

Line 293: What it mean "clinical-radiologic stage"?

Author Response

Thank you for taking the time to review our manuscript.

The attached file addresses your concerns point-by-point

Round 2

Reviewer 1 Report

The authors reviewed the paper improving its quality based on reviewer's suggestion